# Green Tea Catechins and Skin Health

**DOI:** 10.3390/antiox13121506

**Published:** 2024-12-10

**Authors:** Xin-Qiang Zheng, Xue-Han Zhang, Han-Qing Gao, Lan-Ying Huang, Jing-Jing Ye, Jian-Hui Ye, Jian-Liang Lu, Shi-Cheng Ma, Yue-Rong Liang

**Affiliations:** 1Tea Research Institute, Zhejiang University, #866, Yuhangtang Road, Hangzhou 310058, China; xqzheng@zju.edu.cn (X.-Q.Z.); 12216079@zju.edu.cn (X.-H.Z.); 22316139@zju.edu.cn (H.-Q.G.); lanyingh@zju.edu.cn (L.-Y.H.); 12116074@zju.edu.cn (J.-J.Y.); jianhuiye@zju.edu.cn (J.-H.Y.); jllu@zju.edu.cn (J.-L.L.); 2Wuzhou Liubao Tea Research Association, #18, Sanlong Avenue, Changzhou District, Wuzhou 543001, China; zjumasc@aliyun.com

**Keywords:** *Camellia sinensis*, skin care, ultraviolet radiation, antioxidation, anti-inflammation, anti-carcinogenesis, angiogenesis, signaling pathway

## Abstract

Green tea catechins (GTCs) are a group of bioactive polyphenolic compounds found in fresh tea leaves (*Camellia sinensis* (L.) O. Kuntze). They have garnered significant attention due to their diverse health benefits and potential therapeutic applications, including as antioxidant and sunscreen agents. Human skin serves as the primary barrier against various external aggressors, including pathogens, pollutants, and harmful ultraviolet radiation (UVR). Skin aging is a complex biological process influenced by intrinsic factors such as genetics and hormonal changes, as well as extrinsic factors like environmental stressors, among which UVR plays a pivotal role in accelerating skin aging and contributing to various dermatological conditions. Research has demonstrated that GTCs possess potent antioxidant properties that help neutralize free radicals generated by oxidative stress. This action not only mitigates cellular damage but also supports the repair mechanisms inherent in human skin. Furthermore, GTCs exhibit anti-carcinogenic effects by inhibiting pathways involved in tumor promotion and progression. GTCs have been shown to exert anti-inflammatory effects through modulation of inflammatory signaling pathways. Chronic inflammation is known to contribute significantly to both premature aging and various dermatological diseases such as psoriasis or eczema. By regulating these pathways effectively, GTCs may alleviate symptoms associated with inflammatory conditions. GTCs can enhance wound healing processes by stimulating angiogenesis. They also facilitate DNA repair mechanisms within dermal fibroblasts exposed to damaging agents. The photoprotective properties attributed to GTCs further underscore their relevance in skincare formulations aimed at preventing sun-induced damage. Their ability to screen UV light helps shield underlying tissues from harmful rays. This review paper aims to comprehensively examine the beneficial effects of GTCs on skin health through an analysis encompassing in vivo and in vitro studies alongside insights into molecular mechanisms underpinning these effects. Such knowledge could pave the way for the development of innovative strategies focused on harnessing natural compounds like GTCs for improved skincare solutions tailored to combat environmental stresses faced by the human epidermis.

## 1. Introduction

The skin is the largest organ of the human body, serving as a complex and dynamic interface between the internal physiological environment and external surroundings. It comprises two primary layers: the dermis and epidermis. The epidermis acts as a protective barrier against environmental aggressors such as pathogens, chemicals, and physical injuries, while the dermis provides structural support through its rich network of collagen and elastin fibers.

Skin aging can be categorized into two distinct types: intrinsic aging and extrinsic aging. Intrinsic aging is primarily influenced by genetic factors that affect cellular processes over time, leading to changes in skin texture, elasticity, and moisture retention. In contrast, extrinsic aging results from environmental influences such as ultraviolet radiation (UVR), pollution, smoking, and lifestyle choices [1]. Among these factors, prolonged exposure to UVR has been identified as one of the most significant contributors to premature skin aging [2]. This exposure induces oxidative stress within skin cells—particularly fibroblasts—which are crucial for maintaining healthy connective tissue. Oxidative stress occurs when there is an imbalance between free radicals produced during metabolic processes or due to environmental exposures and the body’s ability to neutralize them with antioxidants. This condition can trigger apoptosis in various cell types, including fibroblasts; thus, compromising their function leads to diminished collagen production and impaired wound healing capabilities. Consequently, optimal moisture levels, along with a balanced redox environment, are essential for effective skin repair mechanisms following injury or damage [3].

Green tea catechins (GTCs) represent a group of natural flavanol antioxidants derived from green tea leaves. The key components of GTCs include (−)epigallocatechin gallate (EGCG), (−)epicatechin gallate (ECG), (−)epicatechin (EC), and (−)epigallocatechin (EGC), alongside epi-isomers like (+)gallocatechin gallate (GCG), (+)catechin gallate (CG), (+)gallocatechin (GC), and (+)catechin (C) (Figure 1), among which EGCG is the most abundant. Research indicates that GTCs exert multiple physiological actions beneficial for skin health. Their antioxidant properties help mitigate oxidative damage caused by UVR exposure [4], thereby protecting cellular integrity while promoting longevity among vital cell populations like fibroblasts. Additionally, GTCs possess photoprotective qualities that enhance resistance against sun-induced harm by screening harmful UV rays before they penetrate deeper layers of the skin. Furthermore, GTCs exhibit anti-inflammatory effects [5,6], which may alleviate conditions characterized by chronic inflammation such as acne or rosacea while also supporting overall dermatological health through immunomodulatory activities [7]. GTCs have shown promise in anti-carcinogenesis [8], potentially reducing risks associated with certain forms of skin tumors linked directly to UV exposure. Moreover, studies suggest that GTCs play roles in regulating signaling pathways involved in cellular proliferation and differentiation [9]. They also contribute significantly towards DNA protection mechanisms [10], safeguarding genetic material from damage induced by both endogenous sources like reactive oxygen species (ROS) generated during metabolism and exogenous sources, including UVR. Additionally, their influence on angiogenesis regulation is noteworthy [11]; this process facilitates improved blood flow, which is necessary for nutrient delivery during wound healing phases—a critical aspect often compromised with age-related decline in vascular function within tissues. Both topical application methods—such as creams containing concentrated formulations—and oral supplementation strategies utilizing GTCs have been explored extensively regarding their efficacy against signs of skin aging, along with other dermatological disorders prevalent today. Oral supplements are usually used in the form of capsules filled with 180–500 mg purified GTCs [12,13,14]. Topical GTCs or EGCG are usually used in the form of ointments containing 10–15% GTCs or purified EGCG [15].

In recent years, there have been several review papers published on related topics. Di Sotto et al. [16] provided a comprehensive review of the research progress in understanding the effects of oral GTCs on various skin ailments, such as UV-induced erythema, photoaging, antioxidant defense mechanisms, acne, and genodermatosis. Messire et al. [17] focused on reviewing the antioxidant effects of GTCs for skin protection, cosmetics, and dermatological uses. Mita et al. [18] summarized the potential use of catechins in cosmeceuticals by discussing their antioxidant potential in cosmetic formulations and providing an overview of ongoing clinical trials involving catechins in cosmetics. Aljuffali et al. [19] presented a summary highlighting how nanoencapsulation can enhance skin absorption and the therapeutic efficacy of GTCs while also discussing future applications and limitations associated with nanocarriers for topical delivery. Sinha et al. [20] reviewed the curative potential of EGCG-based nanoforms in wound infection and healing processes by exploring various nano-formulations such as liposomes, lipid nanoparticles, natural polymers, peptide nanostructures, hydrogels, microneedles, nanoparticles, and electrospun fibers used in wound dressing materials. They also discussed future directions for research regarding the contribution of GTCs to clinical studies, along with associated challenges. These reviews primarily focused on clinic studies encompassing specific subjects or cosmetic formulations/methods aimed at improving the skin absorption and therapeutic efficacy of GTCs. However, a comprehensive summarization elucidating the underlying mechanisms responsible for the protective effects of GTCs on skin health is currently lacking and warrants further exploration.

This review aims not only at summarizing existing knowledge surrounding the effects of GTCs in enhancing overall skin health but also at emphasizing the overall mechanism underlying the protective effects of GTCs on skin health, as well as potential applications where these bioactive compounds could serve effectively within skincare products or cosmetic formulations designed specifically to target skin age-related concerns.

## 2. Literature Searching and Screening

The search strategy employed for this study was meticulously designed to retrieve relevant literature spanning from 1 January 2001 to 31 May 2024. This comprehensive review utilized the Web of Science database as its primary source and focused on specific keywords: “tea”, “catechin”, and “skin”. The selection criteria were established to ensure that only pertinent studies were included in the analysis.

During the initial phase of the search process, a total of 407 items were identified. However, upon closer examination, some papers did not meet the inclusion criteria or were not relevant to the topic at hand. Specifically, out of the retrieved documents, 290 papers were excluded based on their content; this group comprised various types of publications, including 68 review articles, 12 entries sourced from meeting proceedings, and 210 items unrelated to our specific inquiry regarding tea catechins and their effects on skin health. After applying these exclusion criteria rigorously, a remaining pool of 117 articles emerged as suitable for further evaluation and were systematically classified into two main categories: ① Clinical trials and epidemiological studies: This category encompasses research involving human subjects aimed at assessing clinical outcomes associated with GTCs consumption or topical and oral applications concerning skin health. ② Physiological function studies: Within this broader category lies a more granular classification into subgroups focusing on distinct physiological actions attributed to GTCs. These subgroups include anti-carcinogenesis studies that investigated how GTCs might inhibit cancer development within skin tissues; anti-inflammation studies that explored mechanisms by which GTCs reduce inflammatory responses linked with various dermatological conditions; wound healing studies that assessed how these compounds facilitate tissue repair processes following injury; antioxidation studies that evaluated antioxidant properties that protect against oxidative stress induced by environmental factors such as UVR; DNA protection studies that analyzed protective effects against DNA damage caused by free radicals or other harmful agents; studies on the regulation of angiogenesis that studied how GTCs influence new blood vessel formation, which is critical for nutrient delivery during healing processes; immunity-boosting studies that examined potential immunomodulatory roles played by catechins in enhancing skin defense mechanisms; photoprotection studies that investigated photoprotective capabilities offered by GTCs against UVR-induced damage; and studies on the regulation of signaling pathways and gene expression that revealed how GTCs interact with cellular signaling pathways affecting gene expression related to skin health. This structured approach not only facilitates a thorough understanding but also highlights diverse aspects through which GTCs can contribute positively towards the maintenance of healthy skin while effectively addressing age-related concerns. By categorizing existing literature in such detail, future researchers can identify gaps in knowledge and areas requiring further investigation within this field.

## 3. Clinical Trials

The strong antioxidant activity of GTCs effectively protects the skin against solar UVR-induced damage, with their functions primarily attributed to total GTCs and certain monomers, notably with EGCG as the most potent component. Both oral administration and topical application of GTCs have been investigated for their photoprotective effects.

The application of a topical formula containing GTCs has demonstrated beneficial effects in protecting human skin during periods of stress. In a randomized controlled clinical trial, 503 patients were allocated to receive either Polyphenon^®^ E (a green tea extract containing GTCs) ointment, Polyphenon^®^ E 15% (GTCs content of 15%, *w*/*w*), Polyphenon^®^ E 10% (GTC content of 10%, *w*/*w*), or a placebo. The topical application was self-administered three times daily on all warts for a duration of 16 weeks. Approximately 53% of patients treated with Polyphenon^®^ E 15% exhibited complete clearance of both baseline and new warts, while the percentage was 51% for Polyphenon^®^ E 10% (*p* = 0.01) and 37% for the placebo group (*p* = 0.03). Notably, women demonstrated a superior response compared to men, with complete clearance achieved by approximately 60% of women and 45% of men in both active treatment groups. Moreover, a wart clearance rate of at least 50% was observed in approximately 78% of all patients treated with either Polyphenon^®^ E ointment at a GTCs concentration of either 15% or 10% [15]. The efficacy of topical EGCG in managing radiation-induced dermatitis was demonstrated in breast cancer patients undergoing radiotherapy. A total of 49 patients who had undergone mastectomy followed by adjuvant radiotherapy were administered daily topical application of EGCG, commencing upon the onset of grade I dermatitis. An aqueous solution of EGCG (660 μM/L) was sprayed three times a day at a dosage of 0.05 mL/cm^2^ on the whole radiation field for two weeks after radiotherapy completion. The maximum observed dermatitis during EGCG treatment was 71.4% with grade 1 toxicity, while 28.6% exhibited grade 2 toxicity and no patients experienced grade 3 or 4 toxicity. Topical application of EGCG resulted in significant pain reduction in 85.7% of patients, alleviated burning sensation in 89.8%, relieved itching in 87.8%, mitigated pulling sensation in 71.4%, and reduced tenderness in 79.6%. These findings suggest that topical administration of EGCG holds promise as a potential therapeutic approach for radiation-induced dermatitis with acceptable levels of adverse effects [21].

A stable water-in-oil emulsion formulation containing 3% ethanolic green tea extract (GTE) rich in GTCs was applied on the cheek skin of healthy human volunteers (*n* = 10, for 8 weeks). Long-term application of this formulation significantly reduced skin sebum production compared to the control cream (*p* < 0.5%) [22]. A non-invasive device with a 2 mm diameter cutometer probe was used to conduct tests on a group of healthy male volunteers (*n* = 10), revealing that the long-term application of cream containing GTE significantly improved the R6 (Uv/Ue) parameter over time [23]. The protective effects of GTCs on skin may be attributed to their antioxidant activity and regulation of gene expression.

Tannase treatment changed the chemical composition of GTE, resulting in changes in its physiological functions. A volume of 40 mL of fresh tea leaf extract was mixed with 10 mg of tannase, then incubated in a water bath at 35 °C for 20 min. The concentration of gallic acid increased significantly from 2.39 ± 0.21 mg/g (dry base) to 17.17 ± 2.45 mg/g, while EGC exhibited an increase from 5.91 ± 1.98 mg/g to 9.38 ± 1.32 mg/g and EC showed an increase from 3.62 ± 0.71 mg/g to 4.95 ± 1.25 mg/g. Consequently, this led to significantly enhanced radical scavenging capabilities [24]. In a six-week randomized, double-blind trial involving four healthy volunteers with noticeable erythema and telangiectasia on their faces, a skin cream containing 2.5% *w*/*w* EGCG was applied twice daily. One side of the face received the EGCG cream, while the other side was treated with a control cream. The results showed that positive staining in the epidermis accounted for 28.4% in sites treated with the control cream compared to only 13.8% in sites treated with EGCG (*p* < 0.001). EGCG treatment resulted in a reduction in HIF-1α expression. Similarly, the EGCG group exhibited a significant decrease in vascular endothelial growth factor (VEGF) expression (*p* < 0.005). Topical application of EGCG demonstrated its ability to influence the induction of HIF-1α and the expression of VEGF, suggesting its potential as a preventive agent for telangiectasias [25].

GTCs can be taken orally and have been found to protect the skin from inflammation caused by sunburn. A study involving 11 participants who orally took three gelatine capsules containing 180 mg GTCs and 25 mg vitamin C each twice daily for 12 weeks revealed that various GTC compounds and their metabolites were present in skin biopsies, blister fluid, and plasma after supplementation. In total, 26 different GTC metabolites were detected in the post-supplementation treatment, with 15 being present in both blister fluid and plasma [13]. A study of female volunteers with dry and sensitive skin revealed that the oral intake of GTCs (47 mg/d) for a duration of 6 weeks, either in fermented milk or alone (*n* = 72), resulted in a decrease in transepidermal water loss and an enhancement in the function of the stratum corneum barrier [26]. During a 12-week study, female volunteers were given a daily beverage containing 1402 mg of GTCs. These participants had specific areas on their skin exposed to UVR from a solar simulator at a 1.25 × minimal erythemal dose (MED). The results indicated a significant reduction in UVR-induced erythema, while blood flow in the exposed skin’s cutaneous and subcutaneous tissues increased. Additionally, improvements were observed in skin elasticity, roughness, scaling, density, and moisture homeostasis [27,28]. The involvement of GTCs in skin metabolism has been found to contribute to their protective effects on the skin. A study conducted on 16 healthy individuals with phototype I/II demonstrated that oral administration of GTCs before and after exposure to UVR resulted in a reduction in erythema response. Following 12-week daily GTCs supplementation (540 mg) combined with vitamin C (50 mg), an increase in benzoic acid levels was observed in skin fluid (*p* = 0.03). Additionally, various intact catechins, methylated gallic acid, and hydroxyphenyl valerolactones were detected both in skin tissue and fluid. The consumption of GTCs led to a reduction in UVR-induced levels of 12-hydroxyeicosatetraenoic acid (12-HETE), indicating that incorporating catechin metabolites into human skin through GTCs intake can prevent increases in UVR-induced 12-HETE [12]. In an experiment, fifty individuals between the ages of eighteen and sixty-five with Fitzpatrick skin types I–II were randomly assigned to receive either a combination treatment consisting of twice-daily doses of 540 mg GTCs and 50 mg vitamin C or a placebo for 12 weeks. The findings demonstrated that oral administration of GTCs prevented declines in fibrillin-rich microfibrils, fibulin-2, and fibulin-5 caused by exposure to solar-simulated UVR (3×MED dose) [29]. These effects are believed to contribute to safeguarding against sunburn inflammation induced by UVR, as well as potential long-term damage mediated by UVR.

However, inconsistent findings have also been noted. The administration of EGCG-enriched Polyphenon E or pure EGCG orally at a dosage of 800 mg/d for a duration of 4 weeks did not demonstrate any protective effects against UV-induced erythema [30]. Similarly, the oral consumption of GTCs (1080 mg/d), along with vitamin C, over a period of 3 months by healthy individuals aged between 18 and 65 years (phototypes I–II) did not exhibit any skin protection against UVR-induced inflammatory challenges such as erythema and leukocyte infiltration when compared to a placebo [5]. A clinical trial was conducted on 17 patients with recessive dystrophic epidermolysis bullosa using a multicenter, randomized, crossover, double-blind, placebo-controlled approach. The results indicated that oral consumption of EGCG at doses ranging from 400 to 800 mg/d for a duration of 4 months did not demonstrate any significant improvement compared to the placebo group [31]. In another study involving healthy White adults (13 males and 37 females aged between 18 and 65 years with Fitzpatrick skin phototypes I and II), a double-blind, randomized, placebo-controlled trial was conducted. It revealed that oral intake of GTCs at a dosage equivalent to consuming 5 cups of tea per day (1080 mg) did not serve as an effective substitute for topical sunscreen [32] (Table 1).

## 4. In Vivo and In Vitro Studies

### 4.1. Antiproliferative Effects on Skin Cancer Cells

The alteration of global climate, particularly due to the depletion of stratospheric ozone, results in an increase in environmental UVR, which is a significant contributor to skin damage and tumorigenesis. Skin cancer comprises three common types: basal cell carcinoma, squamous cell carcinoma, and malignant melanoma [33,34]. It has become crucial to find effective measures for preventing the development of skin cancer.

GTCs have demonstrated their ability to prevent the formation of tumors, including those on the skin. Among these compounds, EGCG has shown the highest efficacy in inhibiting cancer cell growth, metastasis, angiogenesis, and other processes involved in cancer progression. Treatment with EGCG has been found to effectively suppress melanoma cell growth [35]. Both EGCG and ECG have been observed to inhibit the proliferation of A375 human melanoma skin cells; however, ECG exhibited a stronger inhibitory effect compared to EGCG [36]. The administration of tea polyphenols through gavage exhibited inhibitory effects on the proliferation, migration, and invasion abilities of melanoma cells. These effects were found to be dependent on both dosage and time [37]. The growth cycle arrest or induction of apoptosis in tumor cells by GTCs was observed due to their impact on various signaling and metabolic pathways within these cells [34]. In order to demonstrate the protective properties of GTCs against skin cancer, a combination of GTCs and milk was administered to 44 healthy volunteers. This treatment resulted in decreased levels of oxidative stress, improved skin texture, and enhanced integrity among the participants [38]. However, it was discovered that at concentrations below 50 µM, EGCG exhibited inhibitory effects on the clonogenic capability of A431 cells but did not demonstrate any antiproliferative abilities [39]. Photodynamic therapy (PDT) is commonly used for non-melanoma treatment. When EGCG was combined with PDT, it was observed that EGCG enhanced the cytotoxicity of PDT and increased the levels of PS protoporphyrin IX and reactive oxygen species in the A431 cell line, consequently inhibiting its cloning ability. Furthermore, EGCG reduced cell viability while promoting cell death in human skin cancer cell lines A431 and SCC13 [40]. Polyphenon E (PE), also known as sinecatechins, demonstrated significant dose-dependent reductions in cSCC cell proliferation (20–30%) and cell viability (4–21% compared to controls) when administered at 200 μg/mL for 48 h. Interestingly, N-acetyl cysteine, an antioxidant compound, augmented the impact of PE on cell viability [41].

Conventional treatment of skin tumors often leads to drug resistance and toxic side effects. However, the anti-carcinogenic properties of EGCG make it a promising candidate for the development of new and highly effective anti-tumor drugs with fewer side effects. Unfortunately, the clinical application of EGCG is limited due to its poor bioavailability. To address this issue, nanotechnology and co-administration have been introduced as strategies to enhance the bioavailability of EGCG [42]. A study discovered that three types of co-polymerized nano-EGCG, which were prepared using UV blockers like ZnO and antioxidants such as lycopene or olive oil, exhibited stronger inhibitory effects on the growth of melanoma cells and tumor angiogenesis than free EGCG [43]. Mixed lipid–gelatin–EGCG nanoparticles were synthesized using a one-step double-milk method, exhibiting excellent efficiency in encapsulation, drug loading capacity, and controlled release properties. Moreover, these nanoparticles demonstrated remarkable anti-tumor effects both in vitro and in vivo [44]. The combination of EGCG and 3-deazaneplanocin A (DZNep) exhibited distinct cell morphology changes while significantly reducing SCC-13 cell survival. Interestingly, the co-administration of these compounds resulted in an even greater reduction in cell survival compared to their individual treatments [45]. It was observed that the growth and migration of a mouse melanoma cell line (B16F10) were significantly inhibited when EGCG and metformin were combined [8]. The combination of EGCG and diallyl trisulfide (DATS) showed a remarkable synergistic effect in reducing the migration of skin cancer cell line A431 [46]. Sequential administration of EGCG and okadaic acid effectively halted the progression of skin tumors in mice [47]. The presence of silver nanoparticles (AgNPs) enhanced the stabilization of antioxidants such as EGCG. It was observed that EGCG AgNPs exhibited greater cytotoxicity towards human skin melanoma (COLO 679) cells and murine B16 melanoma cells (B16-F0) compared to gallic acid AgNPs and caffeine AgNPs. This suggests that a combination of antioxidants can potentially yield AgNPs with distinct toxicity levels [48] (Figure 2). These findings highlight the promising role of EGCG in both preventing and treating skin cancer, offering renewed hope for future advancements in cancer therapy through further research and technological innovations.

### 4.2. Anti-Inflammation and Wound Healing

The anti-inflammatory and wound healing properties of GTCs were reaffirmed by in vitro experiments. The initiation of wound repair heavily relies on pro-inflammatory reactions, but prolonged inflammation can have detrimental effects on the closure of skin wounds. Studies have shown that therapies based on mesenchymal stem cells (MSCs) possess the ability to activate a series of coordinated cellular processes [49]. By inhibiting JAK2, EGCG treatment effectively hindered the migration of lymphocytes towards epidermal melanocytes [50]. Fluorescein isothiocyanate conjugated EGCG (FITC-EGCG) was observed to enter the cytoplasm and migrate into the nucleus of neonatal human dermal fibroblasts (nHDFs). The presence of a low concentration of EGCG (200 mμ mol/L) resulted in the downregulation of genes associated with cell cycle progression, such as cyclins A/B and cyclin-dependent kinase 1. Consequently, there was a decrease in the proportion of cells in the S and G(2)/M phases of the cell cycle, accompanied by an increase in cells residing in the G(0)/G(1) phase. On the other hand, medium (400 mμ mol/L) and high doses (800 mμ mol/L) of EGCG were found to induce apoptosis [51]. Both a combination of EGCG and α-lipoic acid (ALA) known as EA and gold nanoparticles (AuNPs) combined with EGCG and ALA referred to as AuEA demonstrated significant enhancement in the proliferation and migration of Hs68 and HaCaT cells. The topical application of AuEA on mouse skin expedited wound healing, accompanied by a notable increase in the protein expression of VEGF and angiopoietin-1 [52]. Treatment involving microparticles loaded with EGCG and asiaticoside measuring 10 nm in size effectively suppressed the expression of inflammatory factors such as TNF-α, IL-1 β, and IL-6 in fibroblasts stimulated by lipopolysaccharide [53]. Treatment with MCGE, a derivative of EGCG extracted from Anhua dark tea, demonstrated effective suppression of STAT1 activation and inflammatory cytokines in UVB-exposed HaCaT cells. Additionally, it activated the Nrf2 pathway to prevent ROS accumulation, indicating its potential as a photoprotective agent against UVB-induced inflammation [54]. In skin inflammation, keratinocytes produce VEGF and CXCL-8/IL-8, which play crucial roles. EGCG exhibited significant reductions in the secretion of VEGF and CXCL8/IL-8 by TNF α-stimulated normal human keratinocytes (NHKs) [55].

The beneficial effects of GTCs in reducing inflammation and promoting wound healing were also observed in experiments conducted on living organisms. In vivo tests demonstrated that the application of ECG (200 mμ M) resulted in wound healing without causing any local irritation or inflammation [56]. Furthermore, when mice were treated with electrospun membranes containing 1% EGCG (1EGCG/ poly (lactic-co-glycolic acid, PLGA) membrane), there was a significant increase in cell infiltration and enhanced immunoreactivity of Ki-67 (indicating re-epithelialization at the wound site) and CD 31 (suggesting blood vessel formation) compared to mice treated with PLGA membrane alone [57]. Oral administration of EGCG at a dosage of 10 mg/kg enhanced the closure of skin wounds induced by MSCs in a rat model. This was accompanied by an increase in epidermal thickness. MSCs reduced the expression of pro-inflammatory cytokines such as TNF-α, IL-1 β, and IL-6 in the wound area. The co-administration of EGCG further potentiated this downregulation, indicating a synergistic effect between EGCG and MSCs in promoting skin wound healing [49]. Encapsulation of EGCG into poly-γ-glutamate (γ-PGA) microneedles loaded with L-ascorbic acid (AA) efficiently facilitated the delivery of EGCG into the skin and improved symptoms associated with atopic dermatitis (AD). Weekly application of EGCG/AA-loaded microneedles in mice with AD for four weeks resulted in significant improvements in skin lesions and epidermal hyperplasia. This was achieved by reducing serum IgE and histamine levels, as well as inhibiting the production of IFN-γ and Th2-type cytokines compared to the control group (*p* < 0.05) [6]. Additionally, treatment with EGCG improved collagen through both covalent bonding (C-N bond between lysine of collagen and C2 ring B of EGCG) and non-covalent bonding (hydrogen bond and hydrophobic interaction) within cross-linked collagen hydrogels [58]. Contact dermatitis is a prevalent skin condition, and GTC derivatives show promise as a novel treatment option. Acetyl planar catechin, derived from natural (+)-catechin found in green tea through chemical modification, effectively reduced ear swelling caused by 1-fluoro-2,4-dinitrobenzene (DNFB) and suppressed the expression of inflammatory cytokines such as IL-1β, IL-4, and TNF-α. Additionally, it inhibited the activity of myeloperoxidase in DNFB-treated mice [59]. Diabetic wound healing poses a significant biomedical challenge that requires further investigation despite extensive research conducted over the past few decades. Hyaluronic acids (HA)/PLGA core/shell fiber matrices containing EGCG (HA/PLGA-E) are produced using coaxial electrospinning. The application of HA/PLGA-E matrices significantly reduced the size of streptozotocin-induced wounds in mice, as it enhanced re-epithelialization/neovascularization and promoted increased collagen deposition. This demonstrates the potential use of HA/PLGA-E matrices for the development of strategies to expedite healing and regeneration of diabetic wounds and skin [60].

Psoriasis is a chronic and currently incurable inflammatory dermatological condition characterized by aberrant cell proliferation, dysregulated cellular differentiation, and inflammation, resulting in compromised skin barrier function. In laboratory-cultured keratinocytes, the application of nanoEGCG (a chitosan-based polymeric nanoparticle formulation containing EGCG) demonstrated diminished proliferation and inflammatory responses while promoting cellular differentiation. Application of nanoEGCG on affected areas significantly improved psoriasiform pathological markers (*p* < 0.01) in imiquimod (IMQ)-induced reductions of ear and skin thickness, cell proliferation (Ki-67), redness and scaling, infiltration of immune cells (CD4(+) T cells, macrophages, mast cells, and neutrophils), and blood vessel formation (CD31). The levels of caspase-14 protein expression, early differentiation markers (keratin-10), late differentiation markers (filaggrin and loricrin), and activator protein-1 factor (JunB) were increased. Topical treatment with nanoEGCG demonstrated a more than 20-fold advantage in dosage compared to free EGCG alone [61].

The efficacy of EGCG in promoting wound healing varies depending on the method of application. In a study investigating the impact of EGCG on the viability of perforator flaps, a 4 × 6 cm abdominal skin flap was raised and repositioned before applying EGCG through different methods. The results revealed that both the gavage group (administered 100 mg/kg/d EGCG via a feeding tube) and the intraperitoneal group (injected with 50 mg/kg/d EGCG into the peritoneal cavity) exhibited significantly better outcomes compared to the control group (*p* = 0.03). Conversely, when EGCG was directly injected into the flap at a dosage of 40 mg/kg/d, it resulted in inferior flap contraction and viability (*p* < 0.001). These findings suggest that oral or intraperitoneal administration enhances perforator flap viability, while direct injection diminishes its effectiveness [62] (Figure 3).

### 4.3. Alleviating Oxidative Stress

UVR can cause damage to important macromolecules in the skin, such as proteins and lipids. This can result in sunburn, swelling, excessive cell growth, premature aging, and even skin cancer. One of the main factors contributing to this damage is oxidative stress caused by reactive oxygen species (ROS) [63]. GTCs are abundant in hydroxyl groups, which enable them to act as antioxidants and reducing agents. By doing so, they help protect molecules from oxidative stress. In particular, EGCG, a key type of GTC, is frequently used for its ability to mitigate UVR-induced skin injuries due to its photoprotective properties and anti-aging effects [4,64,65,66]. Earlier research indicated that the application of EGCG on mouse skin had a preventive effect against UVB-induced immunosuppression and oxidative stress [67]. A water-soluble GTE rich in GTCs demonstrated significant improvement in reducing skin wrinkle formation during the process of UV-mediated photoaging in mice. This was achieved by increasing collagen and elastin fiber levels while decreasing the expression of MMP-3 enzymes responsible for collagen degradation. These findings suggest potential anti-wrinkle properties of GTCs [68]. Additionally, EGCG, known for its antioxidant properties, effectively reduced melanin secretion and accumulation in melanoma cells. Consequently, it has been utilized as a cosmetic ingredient with positive effects on both skin hydration and wrinkle formation [69]. However, the instability of EGCG is a significant factor that affects its use in cosmetics. Although GTCs in creams underwent oxidation when exposed to a solar simulator, the decrease in antioxidant capacity of EGCG formulations (21.8%) was lower than the degradation rate (76.9%), indicating that partially oxidized products of EGCG also possess antioxidant properties [70].

EGCG3″Me, a derivative of EGCG found in certain oolong and green tea extracts [71,72], exhibited antioxidant properties by enhancing the vitality of HaCaT cells under oxidative stress caused by H_2_O_2_. This effect was accompanied by an upregulation of heme oxygenase 1 (HO-1) expression. Moreover, EGCG3″Me provided protection against cell death induced by sodium nitroprusside (SNP) while also promoting AKT1-mediated NF-κB activity to enhance cell survival [73].

Excessive generation of ROS has been reported to have an impact on the migration and proliferation of cells responsible for producing keratin, resulting in impaired wound healing among individuals with diabetes [74,75,76]. GTCs have demonstrated efficacy in counteracting the elevation of ROS caused by persistent inflammation observed in chronic wounds. The administration of EGCG via a hydrogel composed of guar gum led to enhanced wound healing in subcutaneous wounds experienced by mice with type 2 diabetes (T2D). Furthermore, an HG-Ag-EGCG hydrogel exhibited remarkable effectiveness in expediting wound healing without scarring through its ability to scavenge ROS [77,78]. Rejuvenation of the wound healing process in diabetic mice was significantly enhanced by administering EGCG on a daily basis. This improvement was achieved through the activation of epidermal keratinocytes, which effectively promoted re-epithelialization of wounds via the K16/NRF2/KEAP1 signaling pathway. The remarkable antioxidant properties and skin-protective effects exhibited by EGCG demonstrate its promising potential as a treatment for chronic wounds in individuals with type 2 diabetes [79].

### 4.4. DNA Protection

Pyrimidine dimers, fragmentation, and methylation occurred results of prolonged exposure to UVR, leading to damage in the DNA structure and injury to the skin [80]. The application of GTCs safeguarded the integrity of skin DNA by preventing damage and facilitating the repair process. Consequently, it exhibited a photoprotective effect on the skin in a dose-dependent manner through various mechanisms [81,82]. Treatment with EGCG resulted in a reduction in single-strand breaks within DNA, unstable sites that are susceptible to alkaline conditions, and mutations in the hypoxanthine–guanine phosphoribosyl transferase (HPRT) gene induced by UVR. This ultimately decreased the likelihood of senescence and apoptosis occurring in human skin fibroblasts (HSFs) [82,83]. The promoter region of genes often contains specific regions known as ‘CpG islands’, which are enriched with double-nucleotide ‘CG’. In normal cells, these CpG islands tend to remain unmethylated; however, tumor cells consistently exhibit methylated forms of DNA. Prolonged exposure to UVR leads to a deficiency of micro ribonucleic acid 29 (miR-29) within skin cells. This deficiency increases susceptibility to both DNA methylation and skin cancer development. By regularly administering EGCG, it is possible to prevent reductions in miR-29 levels. Consequently, this results in decreased levels of DNA methylation and reduced activity from DNA-methyltransferase enzymes. Additionally, there upregulation is observed for tumor suppressor genes p16INK4a and Cip1/p21 expression due to EGCG treatment. Overall, DNA hypomethylation in mouse skin cells could be effectively maintained through EGCG administration. This maintenance strategy has potential benefits in terms of reducing the risk of developing skin cancer [84,85,86].

EGCG also demonstrated its ability to protect DNA at the chromosomal level. It caused a reduction in telomere length and suppressed telomerase activity and decreased DNA stability in tumor cells, effectively hindering carcinogenesis [87]. In normal cells, EGCG inhibited DNA methylation levels within the region where promoter and repressor proteins bind to the telomerase reverse transcriptase gene. This subsequently influenced the expression level and activity of telomerase reverse transcriptase [87,88,89]. EGCG regulated the chromosomal structure to enhance DNA stability. By rearranging chromatin structural units and modifying the PARP1-nucleosome complex’s structure [90], EGCG facilitated DNA repair processes. Furthermore, EGCG-induced interleukin (IL-12) secretion has been reported to mediate nucleotide excision repair for damaged DNA [10,91].

The integrity of mitochondrial DNA can be compromised due to skin damage resulting from factors like photoaging or inflammation triggered by extended exposure to sunlight. The application of GTCs demonstrated a protective effect on UVB-induced damage to mitochondrial DNA, with its effectiveness increasing proportionally with dosage [92]. Furthermore, treatment with EGCG not only prevented the release of damaged mitochondrial DNA into serum but also facilitated its repair process. This intervention led to a significant reduction in serum levels of mitochondrial DNA within burn wounds and contributed to the mitigation of inflammatory responses [93]. Additionally, it was observed that EGCG supplementation upregulated SOD-2 expression, which played a role in safeguarding skin mitochondria against harm caused by ionizing radiation [21] (Figure 4).

GTCs have the potential to modify cancer risk by influencing epigenetic processes that regulate the silencing of tumor suppressor genes through DNA methylation in skin cancer cells. In A431 cells, treatment with EGCG demonstrated a dose-dependent reduction in overall DNA methylation levels and decreased levels of 5-methylcytosine, as well as mRNA and proteins associated with DNMT1, DNMT3a, and DNMT3b. Furthermore, EGCG inhibited histone deacetylase activity and reduced levels of methylated H3-Lys 9 while increasing acetylated lysine 9 and 14 on histone H3 (H3-Lys 9 and 14), as well as acetylated lysine 5, 12, and 16 on histone H4. Additionally, EGCG treatment led to the re-expression of silenced tumor suppressor genes p16 (INK4a) and Cip1/p21 at both the mRNA and protein levels. These findings provide valuable insights into the epigenetic mechanisms underlying potential therapeutic approaches for skin cancer prevention [84].

### 4.5. Promoting Angiogenesis

Angiogenesis is a natural physiological process that involves the formation of new blood vessels from existing ones during early stages of vasculogenesis. It plays a crucial role in normal growth, development, wound healing, and the creation of granulation tissue. However, it also serves as a critical milestone in the progression of tumors from benign to malignant states, prompting the utilization of angiogenesis inhibitors for cancer treatment.

VEGF plays a crucial role in angiogenesis, which involves the formation of new blood vessels [94]. Surgical skin flaps are commonly used but often face the challenge of flap necrosis due to vascular disorders. However, treatment with EGCG has shown promising results in enhancing regional blood flow and promoting the growth of new blood vessels, ultimately improving the survival rate of skin flaps [95]. Additionally, exposure to heat stimulates angiogenesis and triggers inflammatory cell infiltration while also disrupting the dermal extracellular matrix through the activation of matrix metalloproteinases. This process further alters dermal structural proteins and contributes to premature aging of the skin. EGCG effectively suppressed the production of collagenolytic MMP-1 induced by heat by interfering with the pathways of activator protein 1 (AP-1). This interference was observed through the inhibition of heat-induced expression of constituent c-Jun, JunB, and c-Fos proteins. These findings suggest that EGCG has potential as a preventive and therapeutic agent for addressing skin aging caused by heat shock [96]. Abnormal angiogenesis in wound healing has been associated with the receptor for advanced glycation end products (RAGE) [97,98].

Topical application of AuEA was found to expedite skin recovery in mice with diabetes by inhibiting the transcription of RAGE and angiopoietin-2. This treatment also resulted in an increase in VEGF expression, which is regulated by the inflammatory reaction induced by TNFα. In human wound tissue that underwent zonal priming, both VEGFA and CD31 levels were reduced at the transcriptional and protein levels. However, their recovery was enhanced through the use of topical EGCG. The combination of AuNP, EGCG, and ALA significantly accelerated the healing process of diabetic cutaneous wounds by regulating angiogenesis and exerting anti-inflammatory effects [99]. A collagen scaffold modified with a nanocomposite of silver and catechin exhibited angiogenic properties that facilitated scar-free healing in diabetic wounds infected with *Pseudomonas aeruginosa*. This was accompanied by an increase in angiogenesis and a decrease in transforming growth factor-beta 1 (TGF-beta 1) expression, while TGF-beta 3 expression was upregulated [100]. Furthermore, GTCs incorporated into a hydrogel effectively stimulated angiogenesis by enhancing VEGF and CD31 expression. It also reduced inflammatory responses by decreasing IL-6 levels and increasing IL-10 levels, thereby promoting rapid healing of diabetic wounds [101]. Antiangiogenic therapy aims to hinder the formation of new blood vessels in tumors. The growth and survival of endometriosis heavily rely on the development of these blood vessels. EGCG has demonstrated its ability to impede the progression of experimental endometriosis by exerting anti-angiogenic effects [102]. GTCs have been found to reduce inflammatory modulators such as TNF α in human umbilical vein endothelial cells and human aortic smooth muscle cells, as well as to decrease nuclear factor kappa-B levels in vascular smooth muscle cells (VSMCs), both without any adverse effects [103].

The application of EGCG resulted in a decrease in initial blood clot formation following microvascular surgery. Additionally, it led to a significant increase in the average inner diameter when compared to the control group that received saline solution [104]. The use of topical formulations containing GTCs effectively suppressed the expression of genes such as α-SMA, fibronectin, mast cell tryptase, mast cell chymase, TGF-beta 1, CTGF, and PAI-1 when compared to a treated control group. Furthermore, the number of cells positive for mast cell tryptase and chymase decreased noticeably in biopsies from subjects who underwent treatment as opposed to those who did not receive any treatment [105]. TMECG, a compound derived from EC found in green tea, exhibited potent antiproliferative effects against melanoma cells by suppressing the expression of dihydro-folate reductase (DHFR) [106]. The antiangiogenic properties of EGCG were attributed to its ability to inhibit the PI3K/AKT and MEK/ERK signaling pathways [92]. The application of a cream containing EGCG demonstrated anti-angiogenesis activity by modulating VEGF expression, thereby preventing telangiectasias [25] (Figure 5).

### 4.6. Regulating Immune Responses

The protective effects of GTCs against autoimmune diseases have been demonstrated. Studies have indicated that green tea—particularly its active compound, EGCG—can alleviate symptoms and decrease pathological changes in animal models with autoimmune diseases [107]. In a specific case of AD, an experimental microneedle formulation based on poly-γ-glutamate (γ-PGA) was utilized to evaluate the biological activity of various GTCs, revealing that EGCG exhibited the highest level of efficacy. Treatment with MNs loaded with the EGCG/AA vector once weekly for 4 weeks significantly ameliorated skin lesions and modulated immune responses in Nc/Nga mice suffering from AD [6]. The severity of the disease was progressively reduced with increasing doses of dietary EGCG, as it hindered the proliferation of antigen-specific T cells and delayed-type hypersensitivity skin response [108]. Furthermore, it obstructed the differentiation process of pro-inflammatory subpopulations Th1 and Th17, while also preventing IL-6-induced suppression in the development of regulatory T cells. GTCs have been found to enhance both humoral and cell-mediated immunity [7,109,110,111]. Monocytes play a crucial role in the immune system’s defense mechanism. GTCs regulate inflammatory and immune responses by influencing monocyte apoptosis during the negative phase. Experimental tests using EC, EGC, ECG, and EGCG demonstrated that EGCG and ECG could induce monocyte apoptosis. In particular, EGCG exhibited significant potential in promoting monocytic apoptosis and, thus, emerged as a promising novel anti-inflammatory agent [112]. γ-Interferon (IFN-γ) prompts the production of a protein called indoleamine 2,3-dioxygenase (IDO) in various types of immune cells with the aim of boosting evasion from the immune system [9]. During tumor development, IDO expression occurs and hampers T-cell reactions by breaking down tryptophan at a local level. This degradation restrains T-lymphocyte growth while also triggering programmed cell death among these lymphocytes [113]. EGCG significantly inhibited the expression and activity of IDO and suppressed IDO-mediated tumor immune escape by blocking the IFN-γ-triggered JAK-PKC-δ-STAT1 signaling pathway [9,114]. These results indicate that EGCG is a potential immunological and targeted therapeutic agent that can enhance anti-tumor immunity to enhance cancer therapy.

EGCG exhibited specific inhibition of the IFN-γ pathway. In both patients with alopecia areata and healthy controls (HC), EGCG at concentrations of 20–40 µM effectively suppressed the expression of downstream genes involved in the pSTAT1 and IFN-γ pathways, such as HLA-B, HLA-DR, and IRF-1, within cultured HaCaT cells. An in vivo experiment revealed that 20 µM EGCG added to PBMCs from alopecia areata patients significantly reduced CD4+ C119-positive (Th1) cells, as well as CD8+ NKG2D+ subset cells [115]. These results reinforce the role of T lymphocyte-mediated cellular immunity in alopecia areata pathogenesis, representing an important step in the development of EGCG treatments to target these specific lymphocyte subsets.

EGCG was found to prevent the suppression of the immune system caused by UV radiation by inhibiting the infiltration of CD11b+ cells into the skin, reducing IL-10 production in both the skin and draining lymph nodes (DLNs), and increasing levels of IL-12 [10,116]. Additionally, GTPs were observed to enhance the presence of cytotoxic T cells (CD8 cells) responsible for killing tumors, suggesting that this could be a significant mechanism through which GTPs inhibit tumor growth [11]. Melanoma and non-melanoma skin cancers are influenced by UV-induced DNA damage and immunosuppression. In animal models, EGCG has demonstrated its ability to inhibit tumor growth by activating CD8+ T cells. This activation of T cells enhances the immune response against tumors through the suppression of JAK-STAT signaling in melanoma [117] (Figure 6).

### 4.7. Photoprotection

Studies have shown that GTCs provide significant protection against damage caused by UVR when taken orally or topically, suggesting that consuming green tea regularly may help protect skin from the harmful effects of sun exposure. Short-term use of GTE has demonstrated photoprotective effects on the endogenous skin metabolome by mitigating cellular DNA damage induced by UVR stress. By protecting fibulin-5, a protein responsible for maintaining the elasticity and integrity of various tissues in the body, GTCs may help prevent premature aging signs such as wrinkles and sagging skin caused by prolonged sun exposure [81,118]. In a study conducted on mice, it was found that the continuous application of a formulation containing 5% tannase-converted green tea extract (FTGE) for seven days had significant benefits. This treatment effectively prevented the depletion of glutathione (GSH), an important antioxidant in the body. Additionally, it inhibited increases in hydrogen peroxide levels triggered by UVR [24]. An interesting compound found in green tea is GCG, which is chemically more stable than EGCG. When mice were treated with GCG, it was observed that skin pigmentation induced by UVB irradiation was reduced. This suggests that GCG has potential as a natural remedy for hyperpigmentation caused by sun exposure. Furthermore, GCG treatment also showed protective effects against UVB-induced photodamage. It enhanced skin elasticity and collagen fibers, which are essential for maintaining youthful and healthy-looking skin. Moreover, mitochondrial aberrations and melanosome formation were suppressed with GCG treatment [94]. These findings highlight the potential benefits of using GTE such as FTGE and GCG in skincare formulations to protect against oxidative stress caused by UVR exposure.

Human skin fibroblasts (HSFs) and epidermal keratinocytes are two types of cells that make up human skin. These cells play important roles in maintaining the integrity and health of skin. However, they are susceptible to damage caused by UVA irradiation, which is a type of ultraviolet radiation from the sun. It was found that HSFs and epidermal keratinocytes have different sensitivities to UVA irradiation. Incubating these cells with 250 μM EGCG was found to prevent DNA damage induced by UVA exposure [81]. GTCs were also shown to have protective effects against UVA-induced damage. In particular, they mitigated lipid peroxidation (a process that damages cell membranes) and prevented cell death when exposed to UVA alone or in combination with cyamemazine (CMZ) [119]. Without the presence of EGCG or other protective compounds, UVA irradiation led to a decrease in HSFs viability in a dose-dependent manner. The lethal dose for these cells was determined to be 9 J/cm^2^. However, when HSFs were treated with EGCG at a concentration of 10 μg/mL for 2 h prior to UVA exposure, significant improvements were observed. The cellular photodamage induced by UVA was attenuated, resulting in enhanced cell viability. This improvement was accompanied by the increased activity of glutathione peroxidase (GSH-Px), an enzyme involved in antioxidant defense mechanisms within cells. Additionally, levels of superoxide anion radicals and malondialdehyde (MDA), both indicators of oxidative stress and cellular damage caused by reactive oxygen species generated during UV exposure, decreased after treatment with EGCG [120]. These findings suggest that EGCG has potential as a protective agent against UVA-induced damage to human skin cells like HSFs and epidermal keratinocytes (Figure 7).

### 4.8. Regulation of Signaling Pathways and Gene Expression

Signaling refers to the process by which cells communicate with each other, regulating various cellular functions. Immune signaling pathways are intricate networks that encompass diverse receptors, molecules involved in signaling, and factors responsible for transcription. These pathways play a vital role in the body’s capacity to identify and react to pathogens while also maintaining immune balance.

An abnormally high level of cyclooxygenase-2 (COX-2) has been linked to the development of cancer. The administration of EGCG-enriched GTE through oral gavage prior to exposure to tumor promoter 12-O-tetradecanoylphorbol-13-acetate (TPA) effectively suppressed the expression of COX-2 in mouse skin. Similarly, EGCG was found to downregulate COX-2 in human mammary epithelial cells stimulated by TPA. It is known that upstream enzymes such as ERK (extracellular signal-regulated protein kinase) and MAPK (p38 mitogen-activated protein kinase) play a role in regulating COX-2 expression across different cell types. Pretreatment with either EGCG or GTE resulted in the inhibition of ERK activation. Furthermore, EGCG demonstrated inhibitory effects on the catalytic activities of both ERK and p38 MAPK [121]. Melanoma, a highly malignant form of skin cancer, is associated with the overexpression of TRAF6 (TNF Receptor-Associated Factor 6), an E3 ubiquitin ligase that plays a crucial role in signaling transduction. In this study, it was found that EGCG interacted with specific residues of TRAF6, namely Gln54, Gly55, Asp57, ILe72, Cys73, and Lys96. This interaction inhibited the binding between TRAF6 and UBC13(E2), thereby suppressing the activity of TRAF6 as an E3 ubiquitin ligase both in vivo and in vitro. Consequently, treatment with EGCG led to reduced phosphorylation of I kappa Ba and the expression of p-TAK1. Additionally, it prevented the nuclear translocation of p65 and p50 proteins. These effects ultimately resulted in the inhibition of cell growth, as well as migration and invasion abilities in melanoma cells. These findings suggest that EGCG could serve as a promising inhibitor targeting TRAF6 for potential use in chemotherapy or melanoma prevention strategies [122]. EGCG was found to have a suppressive effect on the secretion and accumulation of melanin in melanoma cells [69]. The presence of catechins in GTE led to the inhibition of MiTF and the activation of ERK, resulting in reduced levels of tyrosinase protein, in addition to exhibiting anti-melanogenesis activity [123]. Additionally, EGCG demonstrated a significant reduction in migration induced by epidermal growth factor (EGF), leading to the suppression of EGF-induced phosphorylation of p38 MAPK [124]. EGCG demonstrated the ability to reduce the size of keloid lesions, promote cell death, and inhibit excessive cell growth in keloid disease (KD), a common fibroproliferative disorder with an unknown cause. This was achieved by suppressing PAI-1 activity [125]. Prior treatment with EGCG effectively prevented IFN-γ-induced activation of JAK2 and its downstream signaling molecules, STAT1 and STAT3, in human melanocytes. Additionally, EGCG significantly inhibited the expression of intracellular adhesion molecules (ICAM)-1, CXCL10, and monocyte chemotactic protein (MCP)-1 induced by IFN-γ in human melanocytes. Furthermore, EGCG reduced the levels of corresponding receptors, such as CD11a, CXCR3, and CCR2, on human T lymphocytes. Consequently, this led to a decrease in adhesion between human T cells and melanocytes stimulated by IFN-γ. These findings highlight the potential therapeutic application of EGCG for vitiligo treatment while identifying JAK2 as a promising molecular target [50]. GTCs have exhibited positive outcomes in addressing skin aging, particularly in reducing the appearance of deeper wrinkles and improving elasticity. The process of collagen breakdown, triggered by ROS and pro-inflammatory cytokines, is believed to contribute to these signs of skin aging. In laboratory tests using human dermal fibroblast cells (Hs68 cells) treated with TNF-α, EGCG demonstrated its ability to suppress the expression of the MMP-1 gene. Prior treatment with EGCG at concentrations ranging from 10 to 20 mμ M effectively inhibited the TNF-α-induced secretion and expression of MMP-1. Furthermore, EGCG displayed a reduction in ERK phosphorylation (extracellular signal-regulated kinase), indicating its inhibitory effect on MEK (mitogen-activated protein extracellular kinase) and Src phosphorylation—both upstream signaling proteins within the ERK signaling pathway [126]. EGCG enhanced the expression of genes related to natural moisturizing factors, such as filaggrin (FLG), transglutaminase-1, HAS-1, and HAS-2, while reducing the levels of caspase-8 and -3. This protection provided by EGCG prevented radical-induced apoptosis in HaCaT cells [69]. Theabrownins (TBs), which are oxidized catechins found in black tea, induce apoptosis in melanoma cells through a signaling pathway involving p53/NF-kappa B crosstalk [127]. Caspase-14 is an enzyme involved in epidermal keratinization and plays a crucial role in maintaining periodontal health. By improving cytokeratin localization and increasing the expression of caspase-14 and filaggrin, EGCG promoted both keratinization and protease activity in oral mucosa [128]. EGCG treatment has the potential to safeguard cells against NnV-induced cytotoxicity by reducing the levels of MMP-2 and MMP-9, which are crucial factors in the toxic responses caused by NnV [129]. The topical application of EGCG significantly improved the progression of dermonecrotic lesions triggered by NnV [130]. Alopecia areata is linked to T-lymphocyte dysfunction mediated by IFN and elevated circulating IL-17 levels. In HaCat and Jurkat cells, as well as in peripheral blood mononuclear cells (PBMCs) from alopecia areata patients, a concentration of 40 μM EGCG effectively suppressed STAT1 phosphorylation (p-STAT1). By downregulating JAK2 expression, EGCG targeted p-STAT1 and exhibited a synergistic effect in terms of inhibiting HLA-DR and HLA-B expression through the IFN pathway, thereby preserving immune privilege [115].

UVR plays a significant role in the aging of skin. The application of EGCG resulted in a reduction in UVA-induced damage to the skin and protected dermal collagen from depletion by inhibiting the increase in collagenase mRNA level caused by UV exposure. This was achieved by blocking the binding activities of nuclear transcription factors NF-kappa B and AP-1 [131]. Treatment with EGCG at concentrations below 50 μM restored cell viability in normal human dermal fibroblasts (NHDFs) by up to 83.7% after UVB irradiation compared to a control group [132]. Radiation therapy is commonly used for cancer treatment; however, it can cause severe damage to the skin. Pre-treatment with EGCG significantly improved the survival rate of human skin cells exposed to X-rays and reduced X-ray-induced apoptosis. Additionally, EGCG mitigated mitochondrial damage caused by ionizing radiation by upregulating SOD-2 expression. Prior administration of EGCG before irradiation also decreased ROS levels in HaCaT cells and minimized radiation-induced DNA double-strand breaks, as indicated by reduced formation of γ-H2AX foci. Furthermore, EGCG dose-dependently induced the expression of HO-1, which has cytoprotective properties, through transcriptional activation [21].

The regulation of signaling pathways is also affected by the oxidized form of GTCs. Theasinensin A (TSA), which is derived from EGCG and can be found in black tea and oolong tea, has been observed to inhibit the activities of certain proteins involved in the melanocortin 1 receptor (MC1R) signaling pathway. These proteins include tyrosinase, protein kinase A (PKA), cyclic adenosine monophosphate (cAMP) response element-binding protein (CREB), and microphthalmia-associated transcription factor (MITF). As a result, TSA suppresses melanin synthesis and secretion in B16F10 cells stimulated by α-melanocyte-stimulating hormone (alpha-MSH), as well as NHEMs [133] (Figure 7).

## 5. Conclusions

Clinical and in vivo studies have shown that GTCs can be absorbed through both topical treatment and oral administration, protecting skin from stress-induced aging and diseases such as skin warts, radiation-induced erythema and dermatitis, excessive skin sebum, telangiectasia, and sunburn inflammation.

GTCs act as reducing agents and antioxidants in many reactions due to their abundance of hydroxyl groups, protecting molecules from oxidative damage. GTCs have shown anticarcinogenic effects by affecting multiple signaling and metabolic pathways, leading to growth cycle arrest or the apoptosis of tumor cells. EGCG internalizes into the cytoplasm and has shown anti-inflammatory effects by downregulating IL-1β, IL-4, TNF-α, and AP-1. EGCG also accelerates wound healing by upregulating VEGF and angiopoietin-1 expression. GTCs protect skin DNA from UVR-induced damage and help damaged DNA to repair by reducing DNA methylation by inhibiting the decrease in miR-29, decreasing DNA single-strand breaks, and regulating the chromosomal structure and mitochondrial DNA release. GTCs promote angiogenesis by enhancing VEGF and CD31 expression to accelerate diabetic wound healing and have shown therapeutic antiangiogenic effects by affecting VEGF expression. EGCG suppresses IDO-mediated tumor immune escape by blocking the IFN-γ-triggered JAK-PKC-δ-STAT1 signaling pathway, which is considered to be a potential immunological and targeted therapeutic agent. GTCs have shown photoprotection effects by preventing the UVR-induced depletion of GSH and increases in superoxide anion radicals and MDA. EGCG and GTCs have shown skin-protective effects through the regulation of multiple signaling pathways, i.e., increasing immunity by attenuating IFN-γ-induced phosphorylation of JAK2 and its downstream signal transducer and activator of STAT1 and STAT3 in the JAK-STAT signaling pathway and downregulating COX-2 by inhibiting the phosphorylation of ERK and the activity of MEK in the ERK signaling pathway. The inhibition of the PI3K/AKT and MEK/ERK pathways acts synergistically to enhance the antiangiogenic effects of EGCG through the activation of the FOXO transcription factor (Figure 7).

Despite the promising evidence mentioned above, the application of GTCs in topical formulations is limited due to their thermal instability, particularly in sunscreen products. Moreover, conflicting results have been observed with oral supplementation [5,30,31]. Therefore, further studies are warranted to enhance the thermal stability of GTCs and establish robust clinical evidence regarding the benefits of oral green tea preparations for dermatological conditions, as well as to elucidate safety risks.

## Figures and Tables

**Figure 1 antioxidants-13-01506-f001:**
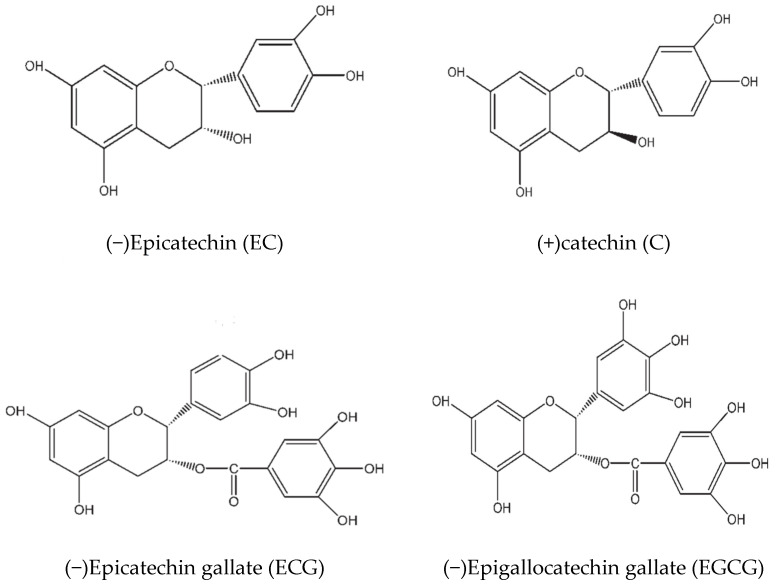
Molecular structure of partial catechins.

**Figure 2 antioxidants-13-01506-f002:**
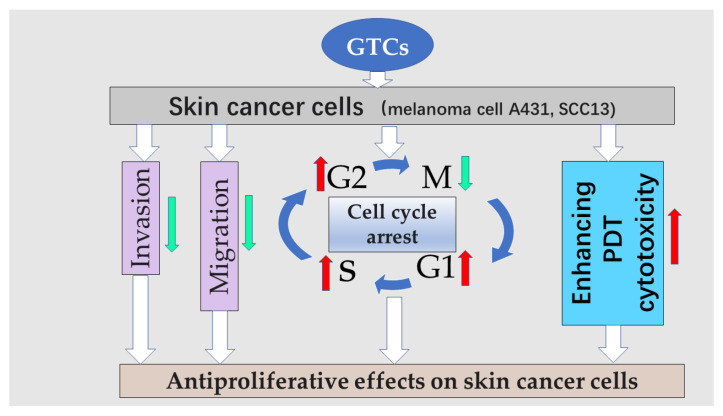
Antiproliferative effects on skin cancer cells. Red arrows indicate increases, and green arrows indicated decreases.

**Figure 3 antioxidants-13-01506-f003:**
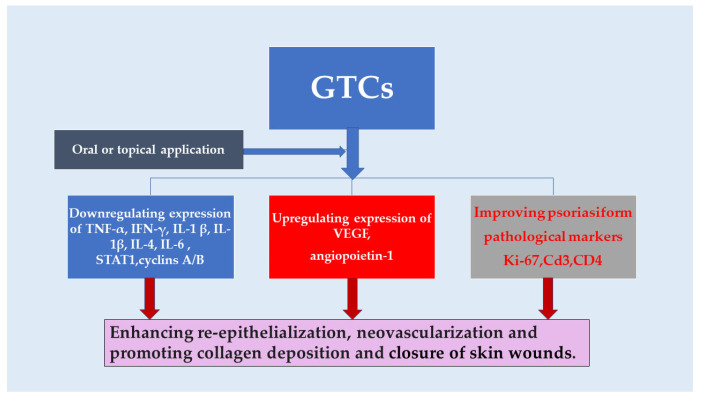
Effects of GTCs on closure of skin wounds.

**Figure 4 antioxidants-13-01506-f004:**
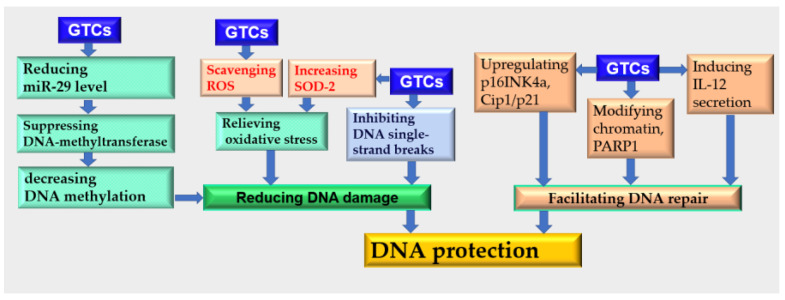
DNA protection effects of GTCs.

**Figure 5 antioxidants-13-01506-f005:**
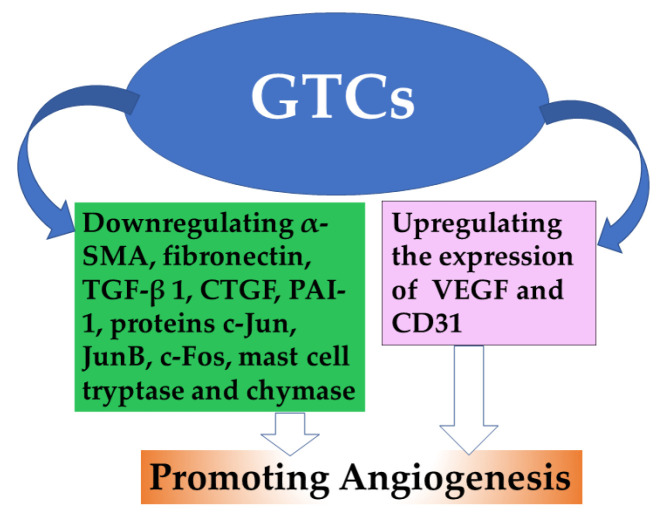
Effects of GTCs in promoting angiogenesis.

**Figure 6 antioxidants-13-01506-f006:**
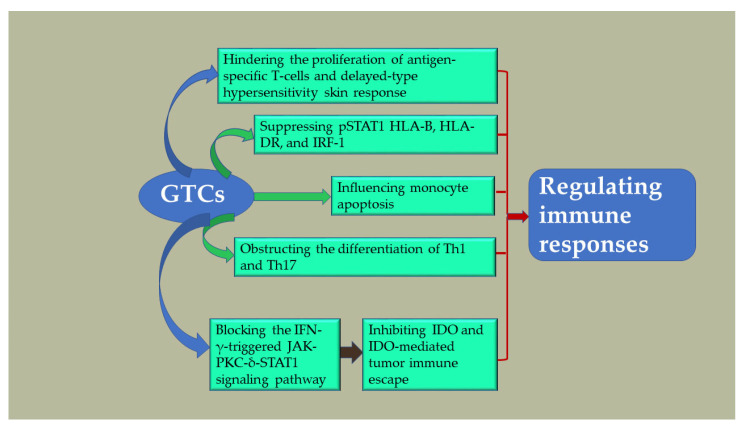
Effects of GTCs in regulating immune responses.

**Figure 7 antioxidants-13-01506-f007:**
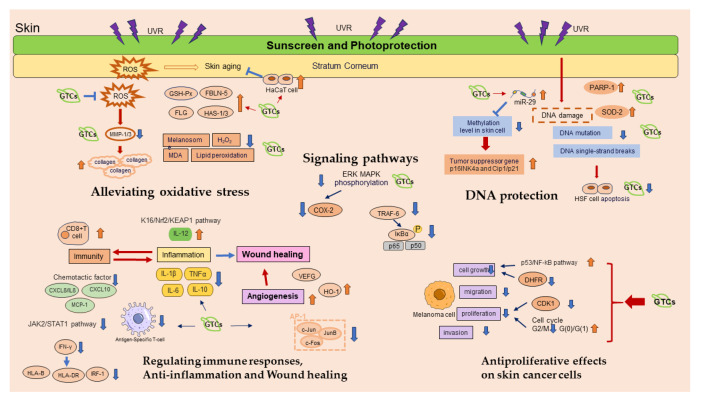
Beneficial effects of GTCs on skin health.

**Table 1 antioxidants-13-01506-t001:** Clinical trials on GTCs and skin health.

No.	Skin Problem	Country	N (Sample Size)	Treatment Dose	Conclusions	Reference
1	UVR-induced erythema and leukocyte infiltration	UK	50 healthy individuals between 18 and 65 years old (phototypes I–II)	Oral intake of GTCs (1080 mg/d) and vitamin C for 3 months	Oral administration of GTCs did not protect against UVR-induced skin inflammation, including erythema and leukocyte infiltration.	[5]
2	UVR-induced erythema	UK	16 healthy White human individuals (males and females, phototype I/II)	Post-supplementation of GTCs (540 mg) plus vitamin C (50 mg) daily for 12 weeks	Oral administration of GTCs before or after UV exposure reduced UVR-induced erythema response.	[12]
3	Sunburn inflammation in skin	UK	12 participants	Consumption of 540 mg GTCs plus 50 mg vitamin C supplementation twice daily for 3 months	GTCs and metabolites were found to be bioavailable in skin.	[13]
4	Anogenital warts	Germany	503 patients	Polyphenon^®^ E (Green tea extract) 15% or 10% ointment or matching vehicle applied topically to all warts 3 times daily for 16 weeks	Topically applying Polyphenon^®^ E ointment was efficient and safe for patients with external genital and perianal warts.	[15]
5	Radiation-induced dermatitis	China	49 patients who underwent mastectomy followed by adjuvant radiotherapy	Topical EGCG daily for 2 weeks	Topical EGCG is a potential treatment for radiation-induced dermatitis with acceptable toxicity.	[21]
6	Skin sebum production	Pakistan	10 healthy male volunteers aged 25–40 years	Continuous use of a stable formulation (water-in-oil emulsion) containing 3% ethanolic green tea extract (GTE) for 8 weeks	Long-term application of the formulation resulted in less skin sebum production than the vehicle control cream.	[22]
7	Skin viscoelasticity	Pakistan	10 healthy male volunteers	Topical use of GTE cream for 60 days	Cream containing green tea extract had definite effects on skin viscoelasticity.	[23]
8	Erythema and telangiectasia	USA	4 healthy volunteers with significant erythema and telangiectasia on their faces	Application of a skin cream containing 2.5% *w*/*w* EGCG on one side of the face and a carrier control cream on the other side twice a day for 6 weeks	Topical EGCG treatments potentially prevented telangiectasias.	[25]
9	Skin barrier	France	72 healthy female volunteers aged 20–45 years	Oral intake of GTCs (47 mg/d) for 6 weeks in fermented milk	GTCs reduced transepidermal water loss and enhanced stratum corneum barrier function.	[26]
10	UVR-induced skin damage	Germany	24 female volunteers aged 18–65 years	Consumption of Cocoa beverage containing 62.51 or 78.96 mg of EC daily for 12 weeks	UVR-induced erythema was significantly reduced, but blood flow to the skin and subcutaneous tissues exposed to UVR increased, with improvement of skin elasticity, roughness, scaling, density, and moisture balance.	[27]
11	UVR-induced skin damage	Germany	60 female volunteers aged 40–65 years	Consumption of beverage containing 1402 mg of GTCs daily for 12 weeks	UVR-induced erythema was significantly reduced, but blood flow to the skin and subcutaneous tissues exposed to UVR increased, with improvement of skin elasticity, roughness, scaling, density, and moisture balance.	[28]
12	UVR-induced sunburn inflammation and damage	UK	50 healthy individuals with Fitzpatrick skin type I–II between 18 and 65 years old	Twice-daily doses of 540 mg GTCs and 50 mg vitamin C or a placebo for 12 weeks	Oral administration of GTCs protected skin against sunburn inflammation induced by UVR and potential longer-term damage mediated by UVR.	[29]
13	UVR-induced erythema	USA	40 healthy males and females ≥ 18 years old with Fitzpatric skin type II or III	Oral administration of EGCG-enriched Polyphenon E or pure EGCG at a dosage of 800 mg/d for a duration of 4 weeks	Oral administration of EGCG or EGCG-enriched productions offered no protection against UVR-induced erythema.	[30]
14	Recessive dystrophic epidermolysis bullosa	France	17 recessive dystrophic epidermolysis bullosa patients with a mean age of 19.4 years	Oral intake of 400–800 mg EGCG daily for 4 months	No significant improvement was observed compared to the placebo group.	[31]
15	Sun protection	UK	Healthy White adults aged 18–65 years with Fitzpatrick skin phototypes I and II (13 males and 37 females)	Oral consumption of 1080 mg GTCs (equivalent to 5 cups tea/d) and 100 mg vitamin C for 12 weeks	Did not serve as a substitute for topical sunscreen.	[32]

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
