# Peer review of "Green Tea Catechins and Skin Health"

_antioxidants, 2024, doi:10.3390/antiox13121506_

Round 1

Reviewer 1 Report

The subject of the research is relevant and could be of great interest to readers.

The main drawback of the manuscript is the lack of systematization of data and information. There are no specific tables showing comparison of existing methods/products. All sections are simply descriptive with no conclusions at the end of the subsections. 

Carefully check as not all abbreviations are explained. 

Other related reviews are published. Please emphasize the novelty of this one.

Author Response

Comment-1:Abstract should be reorganized. After two lines on GTCs the authors are abruptly moving to skin and skin related issues. Line 11: Adding "green" clarifies the sentence. ( ...green tea (Cs) leaves). Introduction is poor. The authors could add a short description on how GTCs are actually used (products type/skin application/oral administration etc...).

Authors’ responses: Thank reviewer-1 for your suggestive comments.

    1) The sentence in Line 11 was changed as: Green tea catechins (GTCs) are a group of bioactive polyphenolic compounds found in fresh tea leaves (Camellia sinensis (L.) O. Kuntze). “Fresh” is used here to avoid repeated use of “Green”.

2) The molecular structures of 4 catechins were presented in Figure 1.

3) In the INTRODUCTION, a short description on how GTCs are actually used was added in red ink. The following related references were cited in the revised version.

4) The final paragraph of the INTRODUCTION provides an overview of recent review papers published on the relevant topics, highlighting the distinctions and advancements presented in this review.

5) The related references were cited in the revised verion.

Rhodes, LE; Darby, G; Massey, KA; Clarke, KA; Dew, TP; Farrar, MD; Bennett, S; Watson, REB; Williamson, G; Nicolaou, A. Oral green tea catechin metabolites are incorporated into human skin and protect against UV radiation-induced cutaneous inflammation in association with reduced production of pro-inflammatory eicosanoid 12-hydroxyeicosatetraenoic acid. British Journal of Nutrition, 2013, 110, 891 – 900. DOI: https://doi.org/10.1017/S0007114512006071

Clarke KA, Dew TP, Watson RE, Farrar MD, Osman JE, Nicolaou A, Rhodes LE, Williamson G. Green tea catechins and their metabolites in human skin before and after exposure to ultraviolet radiation. The Journal of Nutritional Biochemistry. 2016 Jan;27:203-210. DOI: 10.1016/j.jnutbio.2015.09.001. PMID: 26454512; PMCID: PMC4694608.

Lu, P.H.; Hsu, C.H. Does supplementation with green tea extract improve acne in post-adolescent women? A randomized, double-blind, and placebo-controlled clinical trial. Complementary Therapies In Medicine, 2016, 25, 159-163. DOI 10.1016/j.ctim.2016.03.004

Comment-2: The entire manuscript should be organized and presented in a more synthetic manner. Adding tables and figures should make the information more accessible.

Authors’ Responses: Thank reviewer-1 for your suggestive comments.

1) The Tittle Section “3. Clinical Trials and Epidemic Studies” was changed as “3. Clinical Trials”, and the Tittle Section “4. Physiological Function of Tea Catechins on Skin Health” was changed as “4. In vivo and in vitro Studies”. These changes make the two sections look more related logically. And the contents can be precisely reflected in the tittles.

2)Table 1 and Figures were moved to where they first appeared in the context. These changes make the information in the Table and Figures are more easily accessible.

Comment-3:The subject of the research is relevant and could be of great interest to readers. The main drawback of the manuscript is the lack of systematization of data and information. There are no specific tables showing comparison of existing methods/products. All sections are simply descriptive with no conclusions at the end of the subsections. 

Authors’ Responses: Thank reviewer-1 for your suggestive comments.

1) The Tittle Section “3. Clinical Trials and Epidemic Studies” was changed as “3. Clinical Trials”, and the Tittle Section “4. Physiological Function of Tea Catechins on Skin Health” was changed as “4. In vivo and in vitro Studies”. These changes make the two sections look more related logically. And the contents can be precisely reflected in the tittles.

2) The oral supplements used is usually in the form of capsules in which 180-500 mg purified GTCs were filled (Rhodes, LE, 2013; Clarke KA,2015; Lu, P.H 2016). Topically GTCs or purified EGCG are usually used in the form of ointments containing 10-15% of GTCs or purified EGCG (Stockflth et al., 2008). These information was inserted in Lines 86-89 and 140-158. The related references were also added in the references list.

3) Conclusive sentences were added in the end of related subsections.

comment-4: Carefully check as not all abbreviations are explained. 

Authors’ Responses: Thank reviewer-1 for your suggestive comments. All the abbreviations were explained in the context.

Comment-5: Other related reviews are published. Please emphasize the novelty of this one.

Authors’ Responses: Thank reviewer-1 for your suggestive comments. The following information was added in the introduction section.

In recent years, there have been several review papers published on the related topics. Di Sotto et al (2022) provided a comprehensive review of the research progress in understanding the effects of oral GTCs on various skin ailments, such as UV-induced erythema, photoaging, antioxidant defense mechanisms, acne, and genodermatosis. Messire et al (2023) focused on reviewing the antioxidant effects of GTCs for skin protection, cosmetics, and dermatological uses. Di Mita et al (2024) summarized the potential use of catechins in cosmeceuticals by discussing their antioxidant potential in cosmetic formulations and providing an overview of ongoing clinical trials involving catechins in cosmetics. Aljuffali et al (2022) presented a summary highlighting how nanoencapsulation can enhance skin absorption and therapeutic efficacy of GTCs while also discussing future applications and limitations associated with nanocarriers for topical delivery. Sinha et al (2024) reviewed the curative potential of EGCG-based nanoforms in wound infection and healing processes by exploring various nano-formulations such as liposomes, lipid nanoparticles, natural polymers, peptide nanostructures, hydrogels, microneedles, nanoparticles, and electro-spun fibers used in wound dressing materials. They also discussed future directions for research regarding the contribution of GTCs to clinical studies along with associated challenges. These reviews primarily focus on clinic studies encompassing specific subjects or cosmetic formulations/methods aimed at improving skin absorption and therapeutic efficacy of GTCs. However, a comprehensive summarization elucidating the underlying mechanisms responsible for the protective effects of GTCs on skin health is currently lacking and warrants further exploration.

Di Sotto, A.; Gullì, M.; Percaccio, E.; Vitalone, A.; Mazzanti, G.; Di Giacomo, S. Efficacy and safety of oral green tea preparations in skin ailments: a systematic review of clinical studies. Nutrient, 2022, 14, 3149. DOI10.3390/nu14153149

Aljuffali, I.A.; Lin, C.H.; Yang, S.C.; Alalaiwe, A.; Fang, J.Y. Nanoencapsulation of tea catechins for enhancing skin absorption and therapeutic efficacy. AAPS Pharmscitech, 2022, 23, 187. DOI10.1208/s12249-022-02344-3

Messire, G . Serreau, R.; Berteina-Raboin, S. Antioxidant effects of catechins (EGCG), andrographolide, and curcuminoids compounds for skin protection, cosmetics, and dermatological uses: An update. Antioxidants, 2023, 12, 1317. DOI 10.3390/antiox12071317

Mita, S.R.; Husni, P.; Putriana, N.A.; Maharani, R.; Hendrawan, R.P.; Dewi, D.A. A recent update on the potential use of catechins in cosmeceuticals. Cosmetics, 2024, 11, 23. DOI 10.3390/cosmetics11010023.

Sinha, S.; Pant, K.; Mishra, A.; Anand, J. Curative potential of EGCG based nanoforms in wound infection and wound healing. International Journal Of Polymeric Materials And Polymeric Biomaterials, 2024, 1–18. DOI 10.1080/00914037.2024.2398550

Reviewer 2 Report

The manuscript presents a lot of scientific data  concerning the green tea catechins and their effects on skin health.

The authors consulted a very large volume of scientific articles from the specialized literature (127 references) and they systematized the most important aspects related to the effects of catechins from green tea in different pathologies that can occur at the skin level.

The green tea extracts are widely used both for food purpose and for the therapeutic potential of the active compounds in the extracts both alckaloids or polyphenols. The correct medicinal utilization of the active principles from these extracts depends on many factors, and in this context, a systematization of the results obtained by others in this subject is welcome and useful.

Since the volume of information is very large, a better organization of the material would be helpful for the reader.

In this context, my observations and recommendations are as follows:

-          Line 60-61 – more botanical and chemical details could be presented: the scientific name of the species (Camelia sinensis (L), Theaceae), the chemical structure of the more important catechins (EGCG)

-          3. Clinical Trials and Epidemic Studies.

o   What does the expression “ epidemic study” refer to?

o   The chapter could be better structured, depending of the aim of each study

o   Line 133 – the aim of the study is not clear; the Polyphenon E could be described concerning the composition

o   Line 147 – it is not clear what kind of product was used topically, what EGCG was incorpored into, in what proportion ....

o   Line 165-166 – „Tannase treatment resulted in a significant increase in the levels of EGC, EC and gal-165 lic acid (GA) in GTE  more explanation is needed

o   Line 178-179 – it would be useful to have more details about the dosage (amount administered, period)

-          4. Physiological Function of Tea Catechins on Skin Health

o   I recommend revising the title of chapter 4

o   4.1. Anticarcinogenesis. 4.5. Angiogenesis. 4.6. Enhancement of immunity - I recommend revising the titles according the content

-          5. Conclusions –

o   The conclusions should also include comments of the authors regarding the results reported by other researchers, useful information for further research approaches.

Author Response

Comment-1: Conclusion needs improvements.

Authors’ Responses: Thank reviewer-2 for your suggestive comments. The following sentences were added to the CONCLUSION section.

Despite the promising evidence mentioned above, the application of GTCs in topical formulations is limited due to their thermal instability, particularly in sunscreen products. Moreover, conflicting results have been observed with oral supplementation [5,30,31]. Therefore, further studies are warranted to enhance the thermal stability of GTCs and establish robust clinical evidence regarding the benefits of oral green tea preparations for dermatological conditions, as well as to elucidate safety risks.

Comment-2: Major comments

The manuscript presents a lot of scientific data  concerning the green tea catechins and their effects on skin health.

The authors consulted a very large volume of scientific articles from the specialized literature (127 references) and they systematized the most important aspects related to the effects of catechins from green tea in different pathologies that can occur at the skin level.

The green tea extracts are widely used both for food purpose and for the therapeutic potential of the active compounds in the extracts both alckaloids or polyphenols. The correct medicinal utilization of the active principles from these extracts depends on many factors, and in this context, a systematization of the results obtained by others in this subject is welcome and useful.

Since the volume of information is very large, a better organization of the material would be helpful for the reader.

Authors’ Responses: Thank reviewer-2 for your suggestive comments.

The Tittle Section “3. Clinical Trials and Epidemic Studies” was changed as “3. Clinical Trials”, and the Tittle Section “4. Physiological Function of Tea Catechins on Skin Health” was changed as “4. in vivo and in vitro studies”. These changes make the two sections look more related logically. And the contents can be precisely reflected in the tittles.

Table 1 and Figures were moved to where they first appeared in the context. These changes make the information in the Table and Figures is more easily accessible.

Comment-3: Detail comments

In this context, my observations and recommendations are as follows:

-  Line 60-61 – more botanical and chemical details could be presented: the scientific name of the species (Camelia sinensis (L), Theaceae), the chemical structure of the more important catechins (EGCG)

Authors’ Responses: Thank reviewer-2 for your suggestive comments.

1) The scientific name of tea plant (Camelia sinensis) was changes as : Camellia sinensis (L.) O. Kuntze.

2) Chemical structures of the 4 catechins were added as Figure 1.

Comment-4:  3.Clinical Trials and Epidemic Studies.

o   What does the expression “ epidemic study” refer to?

Authors’ Responses: Thank reviewer-2 for your suggestive comments.

There was no epidemic study data in this manuscript and the subtitle was changed as: 3. Clinical Trials.

Comment-5:  The chapter could be better structured, depending of the aim of each study

Authors’ Responses: Thank reviewer-2 for your suggestive comments.

In order to align subtitle 4 with subtitle 3, the wording of subtitle 4 was changed as "4. In vivo and in vitro studies"

Comment-6:  Line 133 – the aim of the study is not clear; the Polyphenon E could be described concerning the composition.

Authors’ Responses: Thank reviewer-2 for your suggestive comments.

Polyphenon® E is an ointment formula containing green tea extract rich in GTCs. Polyphenon® E 15% contains GTCs 15% (w/w) and Polyphenon® E 10% contains GTCs 10% (w/w). This information was added in the revised version.

Comment-7:  Line 147 – it is not clear what kind of product was used topically, what EGCG was incorpored into, in what proportion ....

Authors’ Responses: Thank reviewer-2 for your suggestive comments.

Aqueous solution of EGCG (660 μmol/L) was sprayed three times a day at a dosage of 0.05 mL/cm2 to the whole radiation field for two weeks after radiation therapy completion. This information was added the revised version.

Comment-8:  Line 165-166 – „Tannase treatment resulted in a significant increase in the levels of EGC, EC and gallic acid (GA) in GTE”  more explanation is needed

Authors’ Responses: Thank reviewer-2 for your suggestive comments.

40 mL of GTE was mixed with 10 mg of tannase and then incubated in a water bath at 35 °C for 20 min. Gallic acid concentration increased from 2.39±0.21 mg/g (dry base) to 17.17±2.45 mg/g, EGC increased from 5.91±1.98 mg/g to 9.38±1.32 mg/g and EC increased from 9.38±1.32 mg/g to 4.95±1.25 mg/g. This information was added in the revised version.

Comment-9:  Line 178-179 – it would be useful to have more details about the dosage (amount administered, period)

Authors’ Responses: Thank reviewer-2 for your suggestive comments.

GTCs supplement was gelatine capsules containing 180 mg GTCs each. Subjects orally administered 3 GTCs capsules twice daily for 12 weeks. This information was added in the revised version.

Comment-10: Physiological Function of Tea Catechins on Skin Health

  • I recommend revising the title of chapter 4

Authors’ Responses: Thank reviewer-2 for your suggestive comments. Revision was made accordingly.

The title of chapter 4 was changed as:4. In vivo and in vitro studies.

Comment-11: 4.1. Anticarcinogenesis. 4.5. Angiogenesis. 4.6. Enhancement of immunity - I recommend revising the titles according the content

Authors’ Responses: Thank reviewer-2 for your suggestive comments. Revision was made accordingly.

4.1. Anticarcinogenesis was changed as: 4.1 Antiproliferative effects on skin cancer cells.

4.3. Antioxidation was changed as: 4.1 Alleviating oxidative stress

4.5. Angiogensis was changed as: 4.5. Promoting Angiogenesis

4.6 Enhancement of immunity was changed as: 4.6. Regulating immune responses

Comment-12: Conclusions –

The conclusions should also include comments of the authors regarding the results reported by other researchers, useful information for further research approaches.

Authors’ Responses: Thank reviewer-2 for your suggestive comments. Revision was made accordingly and the following paragraph of text was added in the CONCLUSION section.

Despite the promising evidence mentioned above, the application of GTCs in topical formulations is limited due to their thermal instability, particularly in sunscreen products. Moreover, conflicting results have been observed with oral supplementation. Therefore, further studies are warranted to enhance the thermal stability of GTCs and establish robust clinical evidence regarding the benefits of oral green tea preparations for dermatological conditions, as well as to elucidate safety risks.

Round 2

Reviewer 1 Report

The manuscript can be accepted for publication. The authors put a lot of effort in improving it.

The manuscript can be accepted for publication. The authors put a lot of effort in improving it.

Reviewer 2 Report

The manuscript was improved

The authors responded to the observations made by reviewers and  they made a series of changes to the initial manuscript, taking into account the reviewers' recommendations. The manuscript has a higher scientific level, it is easier for the reader to follow.